# On the fiscal deficit and economic growth in sub-Saharan Africa: A new evidence from system GMM

**Atnafu Gebremeskel Sore[1], Isubalew Daba Ayana[2]\*, Wondaferahu Mulugeta Demissie[3]**

**1** Addis Ababa University, Addis Ababa, Ethiopia, **2** Department of Economics, Wollega University, Nekemte, Ethiopia, **3** Ethiopian Civil Service University, Addis Ababa, Ethiopia

\* isubalewd@wollegauniversity.edu.et

## Abstract

Using a two-step approach GMM, this study examines the short- and long-term effects of fiscal deficit on the economic growth of 42 Sub-Saharan African nations between 2011 and 2021. The World Development Index, the most reliable source, is where the panel data is taken from. Using the Levin-Lin-Chu and Hadri LM tests for unit root, it was determined that there is no risk of a random walk in the data. The study's findings indicate that while the fiscal deficit has short-term, positive, and significant benefits on the economic growth of SSA countries, it has long-term, negative repercussions. According to the system GMM's results, an increase in the fiscal deficit of SSA countries is linked to a short-term increase in economic growth of 0.036 percent, while an increase in the fiscal deficit of one percentage point is linked to a long-term decline in economic growth of SSA countries of 0.013 percent, holding all other factors constant. The study's findings also showed that the budget deficit has a larger positive short-run coefficient than a negative long-run coefficient. The study also revealed that while real effective exchange rates and inflation short-term hinder economic growth, gross fixed capital creation and real interest rates are the primary drivers of economic expansion. Long-term economic growth in the SSA countries is also found to be positively and significantly impacted by gross fixed capital formation. According to the study, SSA nations should manage their fiscal deficits and, in the long run, provide more funds for gross fixed capital development.

## 1. Introduction

Economic policymakers and academic economists have mostly focused on the relationship between fiscal deficit and economic growth. Since emerging nations faced their worst budget deficits since the global financial crisis of 2008, policymakers have paid close attention to the impact of fiscal deficits on economic growth [1,2]. This captured the interest of policymakers even more once the COVID-19 epidemic broke out. This also holds true in other parts of the world, such as the SSA nations [3]. Covid19 pandemic has brought several economic, social, and financial adverse impacts across the globe and the African economy is severely affected by the global pandemic covid19 [4].

**Funding:** The author(s) received no specific funding for this work.

**Competing interests:** The authors have declared that no competing interests exist.

The recent empirical work of [5] found that African inflation has its deep fiscal deficit root in SSA countries. [6] Concluded that COVID-19 created fiscal tension in both developing and developed countries after it was declared a global threat by the World Health Organization in 2019. The scholarly contribution of [7] observed that fiscal deficit has been a global challenge since the world financial crisis of 2008, over the last eleven years. The findings of the work of [8] revealed that budget deficit has become the major bottleneck of the world economy since the time of global financial crises. [9] Argued that over the last decade, fiscal deficit was considered as one of the major indicators of global financial crises. This was accepted as a universal reality in both developing and developed nations worldwide. A related argument was made by [10], who linked government fragmentation to fiscal deficit.

According to [11,12] the fiscal deficit covers a large portion of GDP in SSA countries. For example, in the year 2021, 5.4 percent of the SSA region's GDP was fiscal deficit while it was reduced to 4.5 percent in 2022. This created a huge fiscal burden combined with the level of SSA countries' external debt. This fact implies the fiscal deficit in SSA countries needs further empirical investigations.

Several studies have been conducted to examine the link between fiscal deficit and economic growth in the recent economic literature. For instance, [13] argued that fiscal deficit is one of the causes of rising public debt and a source of economic vulnerability. [14] Concluded that fiscal deficit adversely affects economic growth if the fiscal deficit is not invested in capital formation. Dick (2022) found that fiscal policy hider the performance of macroeconomic variables and the entire economy. A similar study by [15] found contrarily that fiscal deficit has a positive effect on economic growth.

[1] looked into whether there is a perfect amount of fiscal deficit that boosts an economy's growth. According to the study's findings, lower budget deficits are linked to stronger economic growth, whereas higher fiscal deficits are connected with slower economic growth.

Further, [1] studied the role of deficit financing in West African countries implying that the largest portion of West African economic growth is fiscal deficit dominated. The empirical investigation by [16,17] examined the effect of fiscal deficit in emerging markets of West African countries. The main focus of their study was to understand the role of fiscal deficit on the capitalization of the market. Another study by [18] investigated the effect of budget deficits on the economic growth of SSA using the pool mean group estimation. Their study found that fiscal deficit positively affects the economic growth of SSA countries.

The scientific work of [19] was conducted with the main objective of identifying whether the fiscal deficit is inflationary or not in African countries. Their study concluded that fiscal deficit is found to be inflationary in Africa implying the necessity of fiscal consolidation in the contents of Africa. Recent empirical contributions of [20] linked not only fiscal deficit but also triple deficit and economic growth in SSA African countries. Their study found that budget deficit is one of the binding factors for growth.

Notwithstanding the aforementioned earlier studies, it was unclear how the fiscal imbalance affected economic growth. Furthermore, to the best of our knowledge, the current relationship between fiscal deficits and economic growth has not been properly researched, except for the study of [18], who looked directly at the impact of fiscal deficit in SSA nations. The purpose of this study is to present fresh evidence by utilizing system GMM and updated data to capture the recent association between fiscal deficit and economic growth in SSA.

Because it captures the endogeneity problem and yields more efficient estimators in a shorter panel, the panel GMM is preferable to the PMG. Additionally, system GMM takes instrumental variables into account, whereas PMG estimation does not. Additionally, in situations when the data has huge cross-sections and short times, system GMM permits the

moment conditions to be larger than the parameters to provide more accurate estimates. In contrast, PMG estimation does not offer this benefit.

This article's contribution differs in three ways from that of earlier studies. First, it revises the 2011–2021 study periods for two nations in the Southern Africa region. This demonstrates that the current data used to derive the study period's policy implications is current. Thus, the paper discusses the effects of the fiscal deficit on economic growth in the past few decades. Second, a novel econometric approach is used in the study's estimate procedures.

As a result, this study employs the two-step system generalized method of moment. This indicates how this study differs from earlier ones in terms of the data and methodology used. Thirdly, the study is exceptional since it was conducted during the COVID-19 pandemic's devastating effects on the global economy and at a time when the fiscal deficit burden on the SSA countries is greater than it has ever been.

This is how the remainder of the paper is structured. The pertinent theoretical and empirical literature was examined in Section 2. Section 3 presents the study's methods and data, while Section 4 offers the study's findings and discussions. The study's conclusions and policy implications were finally reported in section 5.

## 2. Literature review

### 2.1. The theoretical literature review

The ideas of Ricardian equivalence date back to the contribution of David Ricardo's contribution to economic ideas in 1817 in his Principle of Political Economy and Taxation, of course, one of the most influential economics books. The major idea of the theory was that there is no difference between lump-sum tax and government bonds in financing government budget deficits. It follows that tax cuts today are offset by the future tax rise and the individuals as a rational entity think of this in advance and this leaves the consumption unaffected [21].

The term dubbed today as Ricardian equivalence is just the funding system of David Ricardo which was later widely applied during challenges faced economy between the 1950s and 1970s. The classical thinking of the Ricardian equivalence was just associated with the world of certainty while Barro introduced uncertainty that leads to tax smoothing just to explain it from the perspective of tax distortion [22].

[23] Found that Ricardian equivalence is the major reason for fiscal difficulties and distortions. This is because it leads to the crowd-out effect in consumption and leads the economy into adverse challenges in the long run. [24] Found that Ricardian equivalence does not hold in SSA countries as consumption changed with the change in government debt and government spending adversely affected the economy of the countries during the period under consideration.

The Goldilocks theory of fiscal policy works in the view of fiscal dynamism. This advocates the presence of a zone dubbed as the Goldilocks zone in which deficit is expected to exist permanently but never high in the area. This theory of fiscal deficit further asserts that the level of interest rate is lower compared to the economic growth rate. This leads to fiscal space and creates a deficit in the long run as observed in Japan [25].

The neoclassical theory of fiscal deficit thus recommends the rise of capital formation and accumulation in addition to the rise in the level of saving in an economy. This theory considers any macroeconomic policy that enhances capital accumulation and level of saving to gradually reduce deficit while policies that do not do this are evil for an economy. The goal of such economies needs to be reducing the level of permanent deficit [26].

On the other hand, the theories on economic growth are also reviewed. Three of them were reviewed based on their relevance to this study. First, the classical theory of economic growth,

the so-called theory of specialization, is based on the returns to scale principle of Adam Smith [27]. Accordingly, this theory advocates economies of scale. The better the economies of scale, the better the growth of a nation [28]. Secondly, neoclassical economic growth theory, the theory that emphasizes supply-side factors, especially labor productivity, is reviewed. It is just a little extension of classical theory [29,30]. Thirdly, endogenous growth theory, suggests that growth is mainly due to internal factors instead of external aspects [31]. The major concern of this theory is that ideas and improvements in innovation are the source of growth. All factors within the system are responsible for growth instead of the activities of agents outside the system [32,33]. This study is in line with this endogenous growth theory. The idea is that economic growth in SSA countries is determined mainly by internal factors.

## 2.2. Empirical literature

This section aims to review empirical works that are very related and relevant to fiscal deficit and economic growth. Literature is reporting mixed findings on the link between fiscal deficit and economic growth. Accordingly, some literature supports that the effect of fiscal deficit is dependent on economic crisis, and instability of the economy. It follows that the effect of fiscal deficit on economic growth is based on the challenges the economy is facing. For instance, [7] found that fiscal deficit deteriorated growth during the periods before the global financial crises for ASEAN countries from the period of 2001 to 2015. This case was changed after the world financial crisis of 2008 as the fiscal deficit enhanced growth in these countries.

Contrary to this, other literature supports that the relationship between fiscal deficit and economic growth is based on the short and the long run. For illustration, [18] Examined the relationship between budget deficit and economic growth in SSA countries for the period of 1991–2018. Their study found that the impact of budget deficit is different in the long and short run. Budget deficit influenced economic growth positively in the short run while it impacted growth negatively in the long run. They conducted their study through a pooled mean group estimation technique for a panel of 20 countries of SSA. This reveals the time in which budget deficit occurs and the effect of budget deficit on economic growth. This is one of the areas on which this study focused as a gap.

Other groups of literature found that the link between fiscal deficit and economic growth is based on the rate of growth of the countries and the size of the budget deficit of the countries. The best example of this is the work of [13], who investigated the link between fiscal deficit and economic growth in South Asian countries to find that India and China record one of the leading economic growth rates regardless of the challenges of the fiscal deficit and the overall economic challenges across the globe.

Their purchasing power parity also increased irrespective of the challenges. Following a similar fashion, [1] studied whether an optimal budget deficit promotes economic growth or not in SSA countries. By employing the panel data from 2000 to 2019 along with 48 countries through the help of panel threshold regression, the study concluded that the size of the budget deficit matters on the linkage between the two variables. It was found that the lower the budget deficit of the sample countries, the better their economic growth. Further, the larger budget deficit hampers the economy of countries. It trails that countries should at least lower their budget deficit. This shows that the effect of the budget deficit and economic growth needs further investigation as the literature is reporting contradicting results.

Other literature reports the positive effect of fiscal deficit on economic growth. For example, [34] found that fiscal deficit positively impacted the economic growth of BRICS countries from the period of 1997 to 2016. Their study also witnessed that there is a bi-directional causality between budget deficit and economic growth. Similarly, the work of [35] Found that fiscal

deficit has a positive relationship with the economic growth for the panel of Eurozone countries during the study periods of 1995 to 2015. Employing the multiple linear regression estimation technique, their study found that long deficits are harmful compared to short deficits in the study area during the period under investigation. It can also create a huge macroeconomic imbalance in an economy. Furthermore, the empirical investigation of [36] Found that fiscal deficit affected the economic growth of the Eurozone from the period 2000 to 2021 using the estimation method of panel GMM. The fiscal deficit interacted with the COVID-19 pandemic was also found to be growth enhancing in the area while the interaction of fiscal deficit with the deficit crises was observed to be growth deteriorating.

Contrarily, some groups of literature reported the adverse effect of fiscal deficit on economic growth. To illustrate, [37] studied the effect of a budget deficit within a triple deficit on the economic growth of SSA countries. His study concluded that budget deficit was the major constraint of economic growth in SSA countries from the period of 1980–2018. The study that collected the panel data from 35 African countries concluded that budget deficit is one of the most important factors hindering growth compared to other deficits in SSA countries. Correspondingly, [38] Found the adverse impact of fiscal deficit on economic growth for South Asian countries for the period of 1980–2014.

Their study also revealed the mixed causality between fiscal deficit and the South Asian economies. Another empirical work that reported the negative effect of fiscal deficit on economic growth is the work of [39] which found that the effect of fiscal deficit varies based on the economy of countries. Their study found that there is a positive and significant impact of fiscal deficit on the economic growth of developed nations while it is found to affect the fiscal deficit of developing countries insignificantly. Their study utilized panel data for 37 countries across the globe from 2001 to 2019 inclusive.

A similar empirical investigation [40,41] found that the impact of growth is negative in both the short and long run for the economy of Asian countries for the period of 1980 to 2015. From the heterogeneous panel data analysis carried out on twenty sample countries in Asia, their study concluded that there is an adverse impact of fiscal deficit on economic growth. [14] Sharma found the adverse effect of fiscal deficit on economic growth from the period 1980 to 2017 for the entire economy of India. They concluded that the fiscal deficit should be invested in gross capital formation in the long run. The purpose is to reduce the fiscal deficit financed growth burden and, in the long run, to benefit from the multiplier effect of investment. Thus, their study confirmed the neoclassical growth theory of investment.

Likewise, the empirical study of [42] found that the linkage between budget deficit and economic growth in West Africa is dependent on the level of financial development of the countries. Their study concluded that the adverse effect of fiscal deficit tends to decrease in countries where financial development is relatively better. Further, their study warned that financial development alone cannot cure the economy unless the long-run burden of fiscal deficits is minimized significantly. From this, it is possible to understand that much literature is also reporting the adverse effect of fiscal deficit on economic growth.

Concerning control variables of the study, numerous studies investigated the link between real effective exchange rates and economic growth. For instance, [43] found that undervaluation has a significant impact on economic growth supporting Rodrik's conclusion. Their study utilized a panel of 93 countries for the period 1990 to 2018 to find that the fundamental equilibrium exchange rate model is more effective compared to the Rodrik approach. Similarly, [44] found that a real effective exchange rate has a positive significant impact on the economic growth of Asian countries during the study period of 1981–2016. From the fully Modified Ordinary Least Squares (FMOLS) model analyzed, their study concluded that the real exchange rate has a significant impact on economic growth even during financial crises.

[45] Found that a real effective exchange rate encourages economic growth through improving capital accumulation. Contrary to this, the study found that there is a negative relationship between exchange rate volatility and economic growth. [46] Found that real appreciation decreases real economic growth while real depreciation raises real economic growth. The contrary finding was obtained by [47,48] who investigated the link between exchange rate undervaluation and economic growth to revisit the previous theories. Tang (2015) found reverse causality that runs from economic growth to real exchange rate for the time series data of China's economy. [49] Found positive and significant effect of undervaluation on the economic growth of South Africa using system generalized method of moment's technique of dynamic panel data estimation for the sectorial growth.

As inflation is one of the fundamental macroeconomic problems, its effect on economic growth is studied by several empirical works. For instance, [50] found that inflation hurts the economy if it surpasses a certain threshold level. From the panel data of 32 countries for the period 1980–2009, their study found that inflation is harmful if it is greater than 5.43% in Asian countries. Similarly, [51,52] found that inflation is one of the major determinants of economic growth in non-BCBS countries. From the panel data of seven countries from 1990–2019, their study suggested that inflation targeting and societal-centered policies need to be implemented.

[53] Found that many literatures agree on the negative effect of inflation on economic growth. From the review carried out on international literature, their study also found that there is unidirectional causality between inflation and economic growth. Similarly, [54] found that much empirical literature supports the adverse impact of inflation on economic growth although there is still a debate regarding the relationship between the two variables.

[55] Found that the linkage between inflation and economic growth exhibits a nonlinear relationship for the Turkish Republic. From the thresh-hold analysis conducted, they obtained that thresh hold level was 7.97% during the study period. It was also a study concluded that inflation below the threshold hold positively contributes to economic growth while inflation hurts the economy above the threshhold. [56] Found that inflation beyond a certain limit is detrimental to Africa too. From the panel data of 47 African countries, their study found that the inflation threshold hold level varies in Africa based on the level of income of Africa's regions. In the whole sample countries of Africa, they found that 6.7% is a threshold hold level while it is 9% percent for middle-income countries of Africa.

The linkage between real interest rate and economic growth has been the center of the debate across empirical works in economics. For instance, [52] found that interest rate is a policy variable in transition economies. From the panel data of 38 countries for the study period of 1996–2015, their study concluded that real interest rates can restrict economies to maintain their transitional level. The study suggested that transition economies need a low level of real interest rate that does not deter the level of economic growth. [57] Found a negative relationship between interest rate and economic growth in the long run from the period of 1993 to 2017 for the economy of the Gambia. Similarly, [58] observed that interest rates hurt the economic growth of Swaziland for the study period of 1980–2016. [59] Found that the level of interest rates in both countries is determined by the level of economic growth of countries. Using the Bayesian method from the period of 1961–2014, their study found that the trend in economic growth in a country matters too clearly to understand the effect of interest rates on economic growth.

Concerning gross fixed capital formation as a control variable, the link between capital and economic growth has been the center of economic discussion since the time of the lord Adam Smith. Since then several empirical investigations have been carried out. Recently, [60] found bidirectional causality between capital formation and economic growth. It follows that better

capital formation leads to better economic growth and the better the economic growth, the more conducive environment will be for the capital formation. Similarly, [61] found that there was a negative impact of gross capital formation on economic growth from the period of 1970 to 2012 for the economy of India.

Contrary to this, [62] found that gross fixed capital formation improves institutions and reduces poverty. Through this channel, it enhances the economic growth of SSA Countries. From the two-step system GMM of 41 SSA countries across the period of 1981–2010, they concluded that attention should be delivered to capital formation in the SSA region.

The empirical study by [51] found that gross capital formation is one of the major determinants of economic growth in SSA Africa. From 34 SSA countries throughout 2001–2019, their study concluded that policymakers need to focus on capital formation. Another similar study by [63] found that causality between capital formation and economic growth is based on the development of the country. In Africa, their study observed unidirectional causality between capital formation and economic growth. Utilizing a panel vector autoregressive model for the data from the period of 1980 to 2018, their study concluded that gross capital formation negatively impacts the economic growth of SSA countries.

## 3. Method and data of the study

### 3.1. Data sources and variables

The panel data of 42 SSA countries for the period of 2001–2021 inclusive for this study was obtained from the World Development Indicator Database of the World Bank Group. This data is reliable for two reasons. First, it is from a well-reputed global institution and dependable data (World Bank, 2021; World Development Indicator, 2021). Secondly, it is preferred due to the wide access to the macroeconomic variables data availability. This study employed real gross domestic product (GDP), Fiscal deficit (FDT), inflation (INF), real effective exchange rate (RER), gross capital formation (GFCF), and real interest rate (RIR) as study variables. The choice of the study variables was reasonable as variables are the major causes of the fiscal deficit. The detailed Descriptions of the study Variables are given in Table 1 below.

Table 1 displays the variables of the study and their corresponding descriptions. Accordingly, six (one dependent and five explanatory) variables were employed in the study. The selection of the study variables was reasonable. We included the study variables in line with

**Table 1.  Descriptions of the study variables and expected sign of coefficients.**

| Variables of the study | Description the variables | Expected sign |
|---|---|---|
| GDP per capita (constant 2015 US$)(RGDP) | GDP per capita is gross domestic product divided by midyear population. Data are in constant 2015 U.S. dollars. | |
| Gross fixed capital formation (% of GDP)(GFCF) | | +/_ |
| Real effective exchange rate (RER) | The real effective exchange rate is the nominal effective exchange rate divided by a price deflator or index of costs. | _ |
| Real interest rate (%)(RIR) | The real interest rate is the lending interest rate adjusted for inflation as measured by the GDP deflator. | + |
| Inflation, consumer prices (annual %)(INF) | Reflects the annual percentage change in the cost to the average consumer of acquiring a basket of goods and services. | _ |
| Fiscal deficit (% of GDP)(FDT) | Tax revenue (% of GDP) less general government final consumption expenditure as % GDP. | _ |

Source: Authors' building.

**Table 2. Sample SSA countries selected for the study.**

| | Sample SSA countries selected for the study | | | | |
|---|---|---|---|---|---|
| S.no. | Countries | S.no. | Countries | S.no. | Countries |
| 1 | Angola | 15 | Ethiopia | 29 | Nigeria |
| 2 | Benin | 16 | Gabon | 30 | Rwanda |
| 3 | Botswana | 17 | Gambia, The | 31 | Sao Tome and Principe |
| 4 | Burkina Faso | 18 | Ghana | 32 | Senegal |
| 5 | Cabo Verde | 19 | Guinea-Bissau | 33 | Seychelles |
| 6 | Burundi | 20 | Kenya | 34 | Sierra Leone |
| 7 | Cameroon | 21 | Lesotho | 35 | South Africa |
| 8 | Central African Republic | 22 | Madagascar | 36 | South Sudan |
| 9 | Comoros | 23 | Mali | 37 | Sudan |
| 10 | Congo, Dem. Rep. | 24 | Mauritania | 38 | Tanzania |
| 11 | Congo, Rep. | 25 | Mauritius | 39 | Togo |
| 12 | Cote d'Ivoire | 26 | Mozambique | 40 | Uganda |
| 13 | Equatorial Guinea | 27 | Namibia | 41 | Zambia |
| 14 | Eswatini | 28 | Niger | 42 | Zimbabwe |

Source: Authors' building.

[1]. However, our study is unique as it introduced fiscal deficit in the economic growth models. Thus, our study follows the endogenous growth model theory [30,32].

Table 2 presents sample countries of SSA selected to study the effect of fiscal deficit on the economic growth of SSA countries. The study incorporated forty-two countries from 2011 to 2021. The countries were selected on data availability. Somalia, Eritrea, Malawi, Liberia, Chad, and Guinea were not elected for the study because of the unavailability of panel data on the study variables. This method of selecting a sample is common and popular in panel data analysis.

## 3.2. Specification of system GMM model

Theoretically, a framework that made the investigation between fiscal deficit and economic growth dates back to the time of Keynes and the entire economic curative activities of the 1930s world great depression. However, the model for this study is specified following the endogenous growth models that modeled economic growth as functions of labor and capital. Our present study specified the models based on the work of [34,64] who modeled economic growth as a function of budget deficit and other macro variables that exacerbate fiscal deficit. However, the present model has improved over their model as it included other additional explanatory variables that worsen the Sub-Saharan African fiscal deficit including imports.

Thus, the functional relationship between SSA countries' fiscal deficit and economic growth is given as follows:

$$RGDP = f(\text{FDT}) \tag{1}$$

Where FDT is the fiscal deficit as a percentage of GDP and RDGP is the real gross domestic product.

Eq (1) presents real gross domestic product as a function of fiscal deficit. It follows that the growth of SSA countries can be modeled as the function of their fiscal deficit.

Fiscal deficit, the variable of interest in this study, is included in the present study following the work of [65,66]. However, the current model has improved over the previous works as it included more appropriate control variables in the context of SSA countries.

Including the control variables, the functional form of the model is written as:

$$RGDP = f(\text{FDT, INF, REER, IR, GFCF}) \tag{2}$$

Where, INF is the inflation, RER is the real effective exchange rate, IR denotes the real interest rate, and GFCF shows gross fixed capital formation.

Eq (2) presents the major variable of the study and the inclusion of four control variables into the model.

The inclusion of the variables in the model is reasonable. Accordingly, inflation is included in the model following the empirical works of [38,51,67] while the real effective exchange rate is incorporated as an explanatory variable following [46]. Further, the inclusion of real interest rate in the model is based on [57] whereas the incorporation of gross fixed capital formation followed [62]. However, our present model has improved over them as it incorporated all the variables together in an economic growth model at the same time.

The System GMM (SGMM) model specified for the estimation is:

$$RGDP_{it} = \eta_1 RGDP_{it-1} + \eta_2 EX_{it} + \eta_3 EX_{it-1} + \varsigma_{it} \tag{3}$$

Where, $RGDP_{it}$ denotes the real GDP of SSA countries and $\eta RGDP_{it-1}$ the lag of the real GDP growth of the SSA region. $EX_{it}$ Shows vectors of all explanatory variables of the study while $EX_{it-1}$ represent the lags of the matrix of the explanatory variables and $\varsigma_{it}$ denotes error term. $\eta_1$, $\eta_2$ and $\eta_3$ denotes coefficients of the lagged dependent variable (RGDP), coefficients of a matrix of all explanatory variables, and their corresponding lags respectively. $i$ and $t$ show the cross sections of (42 SSA countries) and the period (2011–2022) of the study.

Eq (3) shows the system GMM containing the lag of the dependent variable and the matrix of all explanatory variables.

In panel system GMM, the error term is given as:

$$\varsigma_{it} = v_{it} + U_{it} \tag{4}$$

Eq (4) shows the error term of the model which is the sum of unobserved fixed effects ($v_{it}$) and idiosyncratic disturbance term ($U_{it}$).

By including the difference operator in the model it is provided as:

$$\Delta RGDP_{it} = \eta_1 \Delta RGDP_{it-1} + \eta_2 \Delta EX_{it} + \eta_3 \Delta EX_{it-1} + \Delta \varsigma_{it} \tag{5}$$

$$\Delta \varsigma_{it} = \Delta v_{it} + \Delta U_{it} \tag{6}$$

Where $\Delta$ represents the difference operator of the model. All others are defined in Eq (3).

Eq (5) and Eq (6) show the difference operator of the system GMM model and the corresponding difference operator of the error term.

Incorporating all explanatory variables of the study, the initial dynamic system GMM model that is estimated in our study is given as:

$$RGDP_{it} = B_0 + B_1 RGDP_{it-1} + B_2 FDT_{it} + B_3 INF_{it} + B_4 RER_{it} + B_5 RIR_{it} + B_6 GFCF_{it} + \varsigma_{it} \tag{7}$$

Where $RGDP$ a real is gross domestic product at time $t$ and $RGDP_{t-1}$ shows the lag of the dependent variable. $FDT$ Shows the fiscal deficit, which is the main variable, which central focus of this study $INF$ Shows inflation, $RER$ is the real effective exchange rate and $IR$ is an interest rate while GFCF is gross fixed capital formation. $B_0$ are coefficients of the lagged dependent variable while $B_1$, $B_2$, $B_3$, $B_4$, $B_5$ and $B_6$ are the coefficients of the explanatory variables. $\varsigma_{it}$ is the vector of the control variables.

Eq 7 shows the dynamic generalized method of moment (GMM) model which is estimated through the system GMM.

All variables were transformed to the natural logarithm just to simplify and make the interpretation of the model more attractive as it results in elasticity interpretation. It enables to carrying out of interpretation by percentage change. The interpretation Thus, the finally estimated result is:

$$logRGDP_{it} = B_0 + B_1 logRGDP_{it-1} + B_2 logFDT_{it} + B_3 logINF_{it} + B_4 logRER_{it} + B_5 logRIR_{it} + B_6 logGFCF_{it} + \varsigma_{it} \tag{8}$$

Where $log$ shows the natural logarithm for easy and clear interpretability.

Eq 8 presents the final estimate of the study in natural log form. Before transforming the value of a major explanatory variable, fiscal deficit, we faced some negative values of fiscal deficit as a percentage of GDP when we calculated the difference between government revenue and government expenditure. Since the log of the negative number is none in economics, as it is in mathematics, we used the most common technique of transforming to handle the negative values in our fiscal deficit data set. To do this, we added a constant value to our fiscal deficit data set before applying the log transform. This method is preferred as it is popular, easy, and more appropriate for managing a few negative numbers that appeared in the data [68–70].

## 3.3. Justification of the model and estimation technique

The estimation technique is carried out by adopting the dynamic panel data model in line with the renowned work of [71]. This estimation technique was previously recommended by [72]. The dynamic panel data estimation technique model was preferred due to its superiority over the others when the time is less than the number of the cross-section of the panel data. It is further preferred since it is compatible and more appropriate when there is weak instrumentation. This study utilized two-step systems GMM which enables coming up with relatively more robust and more efficient estimation when compared to one-step system GMM. The study controlled the problem of over-identification restriction through the use of the Hansen test following the econometric work of [73]. Further, system GMM treats the problem of second-order serial correlation and heteroskedasticity better than other models.

## 4. Results and discussions

### 4.1. Result of descriptive analysis

Table 3 below presents the characteristics of the study variables through descriptive statistics. The entire observation included in the study is 462 (42 cross-section countries for 11 years).

**Table 3. Results of descriptive statistics of the study (2011–2021).**

| Variables | In shorts | Observation | Minimum | Maximum | Mean | Standard dev. |
|---|---|---|---|---|---|---|
| GDP per capita | logGDP | 462 | 7.757506 | 7.83392 | 7.804562 | .0219786 |
| Gross fixed capital formation | logGFCF | 462 | 2.673954 | 2.879956 | 2.788331 | .067121 |
| Real effective exchange rate | logRER | 462 | 4.586047 | 4.607192 | 4.594574 | .0070481 |
| Real interest rate | logRIR | 462 | 1.676896 | 2.455032 | 1.90993 | .2273935 |
| Fiscal deficit | logFDT | 462 | .4934452 | 2.542717 | 1.580225 | .5582076 |
| Inflation | logINF | 462 | 1.548588 | 3.194944 | 2.272067 | .5194298 |

**Source:** Authors computation from STATA 15 **Note:** all variables are in log form.

This implies that the present study incorporated sufficient observation that enables data analysis and drawing recommendations.

The mean value of GDP per capita (log GDP) of SSA countries during the study period is found to be 7.804562 for the period under consideration. Its minimum value is 7.757506 while the maximum value is 7.83392 reflecting that the GDP per capita growth among SSA countries does not vary across SSA countries. Further, this is confirmed by the corresponding low standard deviation of 0.0219786.

Contrary to this, fiscal deficit (logFDT) as a percentage of GDP in SSA has a mean value of 1.580225 with minimum and maximum values of 0.4934452 and 2.542717 respectively. As confirmed by the large standard deviation of 0.5582076, there is a large fiscal deficit variation across SSA countries. Regarding control variables, the average value of gross fixed capital formation in SSA Africa during 2011–2021 is 2.788331 with the corresponding minimum and maximum values of 2.673954 and 2.879956 respectively. The result depicts a standard deviation of 0.067121 reflecting no gross fixed capital formation disparities across SSA countries.

The mean value of SSA countries' real effective exchange rate was found to be 4.594574 with a standard deviation of 0.0070481 while the average real interest rate is 1.90993 through 0.2273935. This implies that sufficient disparities were not observed in SSA during 2011–2021 study periods. Contrarily, SSA countries' inflation during the study period is averaged to be 2.272067 with huge disparities across panels reflected by 0.5582076.

Table 4 below presents the correlation between the study variables. The result indicated that the variables of the study are related. This implies that studying the magnitude of the effect of fiscal deficit on economic growth is on the right track. For instance, gross fixed capital formation (logGFCF) and real interest rate (logRIR) are positively correlated with the GDP per capita of SSA countries while GDP per capita is negatively correlated with fiscal deficit, real effective exchange rate, and inflation. The correlation coefficient of fiscal deficit as a percentage of GDP (logFDT) and GDP per capita of SSA countries during the study period is negative 0.8146 showing that these two variables are strongly correlated. This confirms the expected sign.

Table 5 above presents lists of sample SSA countries with the highest and lowest fiscal deficit during the period of 2011–2021. Accordingly, Botswana is found to be leading in the fiscal deficit in SSA followed by Ghana, Seychelles, Lesotho, and Cote d'Ivoire ranked second to fifth respectively. The greater fiscal deficit in Botswana might be attributed to the acute deterioration of mining revenue as the country mainly depends on revenue from minerals (Timuno & Eita, 2020). Contrary to this, Tanzania, Congo, Dem. Rep. and Sierra Leone rank 8th, 9th, and 10th correspondingly. Further, Gabon leads SSA countries with the lowest fiscal deficit while Rwanda, Uganda, Mauritius, and Namibia follow respectively. Similarly, Nigeria, Cote d'Ivoire South Africa, and Mozambique rank 7th to 10th respectively. This implies that the leading

**Table 4. The result of the pairwise correlation coefficient of the study variables.**

| Variable | logGDP | logGFCF | logRER | logRIR | logFDT | logINF |
|---|---|---|---|---|---|---|
| **logGDP** | 1.0000 | | | | | |
| **logGFCF** | 0.1986 | 1.0000 | | | | |
| **logRER** | -0.1868 | 0.1758 | 1.0000 | | | |
| **logRIR** | 0.2601 | 0.5541 | 0.5831 | 1.0000 | | |
| **logFDT** | -0.8146 | 0.0860 | 0.2106 | -0.0938 | 1.0000 | |
| **logINF** | -0.3886 | -0.6442 | -0.4610 | -0.7686 | 0.0288 | 1.0000 |

**Source:** Authors computation from STATA 15 **Note:** all variables are in log form.

**Table 5. List of ten SSA countries with highest and lowest fiscal deficit (2011–2021).**

| SSA countries with the highest fiscal deficit | | | SSA countries with the lowest fiscal deficit | | |
|---|---|---|---|---|---|
| Rank | Country | Average Value of FDT | Rank | | Average Value of FDT |
| 1st | Botswana | | 1st | Gabon | |
| 2nd | Ghana | -10.0975 | 2nd | Rwanda | -3.285832 |
| 3rd | Seychelles | | 3rd | Uganda | |
| 4th | Lesotho | | 4th | Mauritius | |
| 5th | Cote d'Ivoire | | 5th | Namibia | |
| 6th | Zimbabwe | | 6th | Guinea-Bissau | |
| 7th | Equatorial Guinea | | 7th | Nigeria | |
| 8th | Tanzania | | 8th | Cote d'Ivoire | |
| 9th | Congo, Dem. Rep. | | 9th | South Africa | |
| 10th | Sierra Leone | | 10th | Mozambique | |

**Source**: Authors' construction from STATA15. FDT is Fiscal deficit.

economies in SSA countries have relatively lower fiscal deficits while the fast-growing economies such as Botswana faced acute fiscal deficits during the period under consideration. The average value of fiscal deficit in the ten countries with the highest fiscal deficit is negative (-10.0975) whereas it is negative (-3.285832) in the list of ten countries with the lowest fiscal deficit in SSA countries. This result also reveals that fiscal deficit is the major challenge even in the leading economies of SSA countries.

## 4.2. Results of panel unit root tests

Two familiar methods of panel unit root tests were employed in the study to prohibit the risk of spurious regression results. The first is the Levin-Lin-Chu unit-root test while the second method utilized is the Hadri LM test for unit root. Both methods have different null and alternative hypotheses to check for unit roots in the panel data. The results of stationary are provided in the following tables.

Table 6 presents the results of the Levin-Lin-Chu unit-root test at the level and the first difference. Accordingly, all variables are found to be stationary at the first difference. However, logFDT and logRIR are found to be non-stationary at level. On the other hand, logINF is found to be stationary at a level with a 10 percent level of significance. The fact that all variables

**Table 6. The results of the Levin-Lin-Chu unit-root test.**

| Levin-Lin-Chu unit-root test at level | | | | Levin-Lin-Chu unit-root test at first difference | | | |
|---|---|---|---|---|---|---|---|
| Variables | Statistics | P-value | Decision at I(0) | Variables | Statistics | P-value | Decision I(1) |
| logGDP | -12.1099 | 0.0000** | Stationary | logGDP | -18.9433 | 0.0000** | Stationary |
| logGFCF | -2.9138 | 0.0018** | Stationary | logIMP | -11.0701 | 0.0000** | Stationary |
| logRER | -12.4230 | 0.0000** | Stationary | logRER | -23.0827 | 0.0000** | Stationary |
| logRIR | 95.8608 | 1.0000 | Non-stationary | logRIR | -8.3373 | 0.0000** | Stationary |
| logFDT | 26.1285 | 1.0000 | Non-stationary | logFDT | -8.6285 | 0.0000** | Stationary |
| logINF | -1.4510 | 0.0734* | Stationary | logINF | -29.0836 | 0.0000** | Stationary |

**Source:** Author's computation from Stata 15.

* Shows significant variables at 10 percent while

** shows significant series at a 1 percent level of significance.

**Table 7. Results of hadri LM test for unit root.**

| Hadri LM test for unit root at the level | | | Hadri LM test for unit root at first difference | | | |
|---|---|---|---|---|---|---|
| Variables | Statistics | P-value | Variables | Statistics | P-value | Decision |
| logGDP | 7.2234 | 0.0000** | logGDP | -6.6272 | 1.0000 | Stationary I(1) |
| logGFCF | 25.2660 | 0.0000** | logIMP | -2.6845 | 0.9964 | Stationary I(1) |
| logRER | -3.3057 | 0.9995 | logER | -6.5434 | 1.0000 | Stationary I(1) |
| logRIR | 6.6408 | 0.0000** | logRIR | -0.7798 | 0.7823 | Stationary I(1) |
| logFDT | -0.1551 | 0.5616 | logFDT | -9.6798 | 1.0000 | Stationary I(1) |
| logINF | 13.8673 | 0.0000** | logINF | -0.1984 | 0.5786 | Stationary I(1) |

**Source:** Author's computation from Stata 15.

**Shows that P value is significant and we cannot reject the alternative hypothesis that says some panels contain a unit root.

are stationary at the first difference is that the data is safe to continue with the analysis and free from spurious regression. The null hypothesis of the Levin-Lin-Chu unit-root test is that all panels contain unit roots while the alternative hypothesis is that panels are stationary.

Table 7 presents the result of the Levin-Lin-Chu unit-root test is confirmed by the Hadri LM test for unit root which indicates better confidence regarding our data quality and safety to proceed with regression. The null hypothesis of the Hadri LM test for unit roots is all panels are stationary with the alternative hypothesis some panels contain unit roots. Accordingly, some (logRER and logFDT) study variables are stationary at a level while all study variables are stationary when converted to the first difference. It follows that the null hypothesis cannot be rejected at first difference showing that the result is similar to that of the Levin-Lin-Chu unit-root test. We are now confident enough that the result of our two-step system GMM is free from the risk of spurious regression.

## 4.3. Results of the Hausman test for model selection

Table 8 presents the Hausman test of model selection result and shows that the coefficient of the lag 1 of the dependent variable in the Pooled OLS model is found to be 1.796424 while that of the fixed effect Model is 0.944916. Further, the coefficient of lag 1 of the dependent variable from the two-step difference GMM is found to be 0.3562105. This implies that 0.3562105 is less than 0.944916 and 1.796424 (0.3562105<0.944916<1.796424). This means system GMM is suitable for this study compared to the difference GMM. Accordingly, system GMM is selected in this study as per the result of the Hausman model selection test.

## 4.5. Results of the short-run

Table 9 below presents the short-run effect of fiscal deficit on the economic growth of SSA from the period of 2011 to 2021. The result of the study reveals that the variable of interest, fiscal deficit as a percentage of GDP (logFDT), has a positive and significant effect on the economic growth of SSA countries in the short run. The result of the two-step system GMM (1) also revealed that a percentage change in fiscal deficit of SSA countries, other things that

**Table 8. Results of the Hausman test of model selection.**

| Lagged GDP | Pooled OLS model | Fixed Effect Model | Difference GMM model(two-step) | Decision |
|---|---|---|---|---|
| **L.logGDP** | 1.796424 | 0.944916 | 0.3562105 | System GMM is more appropriate |

**Source:** Author's computation from STATA 15. L.logGDP shows the first lag of GDP, the dependent variable of the study.

**Table 9. Short-run effect of fiscal deficit on economic growth in SSA (2011–2021).**

| Variables of the study | System GMM result(two-step)(1) | Fixed effect model(2) | Random effect model(3) |
|---|---|---|---|
| L1.logGDP | 0.9996989* (0.0070207) | 0.990724* (0.0728763) | 0.944916 * (0.0770183) |
| logGFCF | 0.648299* (0.0406292) | 0.6025781*** (0.0265254) | 0.6391952** (0.015628) |
| logRER | -0.522711 * (0.0341499) | -0.5088408** (0.0250919) | 0.511821** (0.0412028) |
| logRIR | -0.0745543** (0.0387892) | -0.0742543** (0.03833) | -0.0743407*** (0.013328) |
| logFDT | 0.0365444* (0.0023687) | 0.03562105 * (0.0162567) | 0.03285422* (0.0596802 |
| logINF | -0.0355858* (0.0029494) | -0.3025781* (0.0042565) | 0.0357493* (0.0036234) |
| Constant | -43.3916* | -41.563546 | |
| Model diagnostic results | | | |
| Number of observations | 420 | 420 | 420 |
| Wald chi2(7) Prob > chi2 | 0.000 | | 1032.77 |
| Prob > F | | 0.000 | 0.000 |
| Number of groups | 42 | | |
| Number of instruments | 19 | | |
| F(41, 370) | | 1611.55 | |
| Arellano-Bond test for AR(2) | 0.593 | | |
| Hansen test Prob > chi2 | 0.346 | | |

**Source:** Author's computation from STATAT 15. Economic growth (logGD) is the dependent variable of the study. *,**, *** denote 1%, 5%, and 10% levels of significance respectively, in parenthesis shows the corrected standard errors for system GMM and standard errors for fixed and random effect models.

remain constant, is associated with a 0.036 percent increase in economic growth during the study period. This result is found to be consistent in the three models estimated in Table 5. This implies that fiscal deficit enhances growth in the short run as countries finance their budget through deficit financing. The result of our study is consistent with the findings of [18,40].

The result reveals that the first lag one of GDP (L1.logGDP) is found to positively and significantly affect the economic growth of SSA countries. A one percent change in the first lag one of GDP, holding all other factors constant, is associated with a 0.999 percent change in the economic growth of SSA countries. The three models estimated to check the consistency of the result confirmed this fact. This result reflects the fact that last year's better economic growth enhanced the current year's economic growth.

Regarding control variables of the study, the result from the two-step system GMM revealed that gross fixed capital formation (logGFCF) affects the economic growth of SSA countries positively and significantly in the short run. Accordingly, a percentage change in the gross fixed capital formation of SSA countries, holding all other factors constant, is associated with a 0.648 percent increase in economic growth during 2011–2021. This result remained consistent among the three models estimated to check consistency. This shows that investment in fixed capital formation creates a conducive environment for the economic growth of SSA in the short run. The work is consistent with the findings of [18,62,67].

The result of the two-step system GMM also shows that the other control variable, a real effective exchange rate (logRER), affects the economic growth of SSA negatively and significantly. The coefficient of the real exchange rate is found to be negative implying that a one percent deterioration in the real effective exchange rate in SSA, holding all other things constant,

is associated with a 0.522 percent decline in economic growth. This result is significant at a one percent level in model 1 and significant at 5 percent in models 2 and 3 of Table 5. This effect of real effective exchange rate is consistent across all estimated models. This implies that the deterioration of the real effective exchange rate hurts the economic growth of the region. The result found is consistent with the findings of the recent findings of [44,45,49].

In addition to this, the coefficient of real interest rate (logRIR) is found to have a negative coefficient implying that a one percent change in the rate of real interest is associated with a 0.074 percent decline in economic growth, other things remaining constant. This negative effect of the real interest rate on the economic growth of SSA during the study period is significant at a 5 percent level of significance in models 1 and 2 and significant at a 10 percent level in model 3. It implies that financial development in SSA is very low-slung. This corroborates the findings of [52,57].

Finally, Inflation (logINF) is found to adversely impact the economic growth of SSA countries during the period under investigation. The result of the two-step system GMM revealed that a percentage change in the rate of inflation (consumer price index), all other things kept constant, is associated with a -0.035 decline in economic growth of SSA during 2011–2021. The sign and coefficient of inflation remained inconsistent with columns 1 and 2 in column 3 of Table 5. This implies that the inflation rate is very high (double digit) in SSA countries hurts economic growth. This result corroborates with [54–56].

## 4.6. Results of the model fitness

The bottom section of Table 9 above reveals that the two-step system GMM estimated reveals that the number of groups is greater than the number of instruments (42>19) reflecting that the model is built with a collapse option and it is good and what is desired as the number of instrument is reduced. Further, the estimated model revealed that the Arellano-Bond test for AR (2) estimated is 0.593 and it is insignificant reflecting that the estimated model does not suffer from second-order serial correlation. This implies that our estimated model has passed a test for zero autocorrelation in the first differenced errors. On the other hand, the Hansen test is found to be 0.346 showing there is no problem of over-identification of the estimated model.

## 4.7. Results of the long-run

Table 10 presents the long-run effect of variables on economic growth. The result reveals that fiscal deficit harms the economic growth of SSA countries during the period under consideration. Unlike the short-run result, in the long run, a one percentage change in fiscal deficit, keeping all other things constant, is associated with a 0.013 percent decline in the economic growth of SSA countries over the last decades. The result of the two-step system GMM reveals

**Table 10. Long-run effect of fiscal deficit on economic growth in SSA (2011_2021).**

| Variable | Variables | Coefficients | The GMM long-run generation command |
|---|---|---|---|
| Gross fixed capital formation | logGFCF | 0.4699954* | nlcom (_b [logGFCF])/(1-_b[L.logGDP]) |
| Real effective exchange rate | logRER | -0.1158054 | nlcom (_b [logRER])/(1-_b[L.logGDP]) |
| Real interest rate | logRIR | 0.073553 | nlcom (_b [logRIR])/(1-_b[L.logGDP]) |
| Fiscal deficit | logFDT | -0.0137324* | nlcom (_b [logFDT])/(1-_b[L.logGDP]) |
| Inflation | logINF | -0.0591008 | nlcom (_b [logINF])/(1-_b[L.logGDP]) |

**Source:** Author's computation from STATA 15.

*, **, *** denotes 1%, 5%, and 10% level of significance respectively. All variables are in log form.

that the positive coefficient of fiscal deficit in the short run is greater than the negative coefficient. This implies that although the fiscal deficit can finance growth in the short run, it is a burden in the long run. The result of this study was previously supported by [35,36].

Gross fixed capital formation continues to have positive and significant effects in the long run too. The result reveals that a one percentage change in gross fixed capital formation in the long run, ceteris paribus, is associated with a 0.469 percent improvement in the economic growth of sample SSA countries. However, the coefficient of gross fixed capital formation in the long run is less than that of the short run. This implies that SSA African countries have a huge fiscal deficit that lowers their attention on investing in capital formation. The result of this study was previously supported by [63].

On the other hand, the effect of the real effective exchange rate (logRER) on the economic growth of SSA African countries remained negative in the long run. It is revealed that a one percent change in real effective exchange rate deterioration, keeping all other factors constant, is associated with a 0.115 percent decline in the economic growth of SSA countries during the last decades. However, the negative effect of a real effective exchange rate in the long run is insignificant. Similarly, the effect real interest rate in the long run remained positive but insignificant. The result of the study revealed that a percent change in real interest rate, in the long run, is associated with a 0.073 percent increase in economic growth, keeping all other things constant. This result corroborates the work of [45,52].

Finally, the long-run effect of inflation, in the long run, is found to be positive but insignificant. The long-run coefficient generated from the two-step system GMM depicted that a percentage change in an inflation rate, keeping all other things constant, is associated with a 0.059 percent decrease in the economic growth of SSA countries over the past decade. This shows that inflation is harmful to economic growth in both the short and long run in the SSA countries. This result of our study corroborates with [54].

## 5. Conclusion and recommendation

### 5.1. Conclusion

This study examined the effect of fiscal deficit on the economic growth of 42 sub-Saharan African countries that were selected based on data availability. The study used dynamic panel data spanning from 2011 to 2021 and two-step system GMM estimation techniques to estimate the result. Fiscal deficit as a % of GDP used in the study was calculated as tax revenue as % of GDP less general government final consumption expenditure as % GDP.

From the result of the two-step system GMM, the studies concluded that the effects of fiscal policy in the short run and the long are different. The result of our study confirmed that fiscal deficit has a positive and significant effect on the economic growth of SSA countries in the short run while it adversely and significantly affects economic growth in the long run. This implies that the effects of fiscal deficit on the economic growth of SSA countries vary depending on the time. The short-run effect of fiscal deficit on economic growth in SSA countries is in line with Keynesian theory of fiscal deficit while the long-run result of our study corroborates with the neoclassical theory of fiscal deficit.

The result of the two-step system GMM also shows real effective exchange rate and inflation affect the economic growth of SSA negatively in the short run. Although the negative effect of real effective exchange rate and inflation is significant in the short run, the negative effect of both variables remains insignificant in the long run. Contrary to this, the result of the study revealed that gross fixed capital formation and real interest rate are the engines of economic growth in the SSA region in the short run. While the contribution of gross fixed capital formation to economic growth is significant in the long run, the positive contribution of real interest

rate to economic growth is observed to be insignificant in the long run for the economy of SSA countries.

## 5.2. Recommendation

Thus, the policy implication is that SSA countries should ensure the efficiency of the borrowed funds by investing them in capital projects that can generate future huge returns and enhance gross fixed capital formations. Furthermore, SSA countries should reduce the overall fiscal deficit as it leads to debt accumulation through deficit financing in SSA countries by rising expenditure on developmental projects.

## 5.3. Recommending further studies

Forthcoming inquiries might enlarge the set of indicators to explore the effects of other variables on SSA countries. For that matter, comparisons between different income groups of SSA countries would lead to interesting insights. Third, the period of analysis covered nearly 11 years as it is appropriate with the system GMM. Future investigations could expand the time frame to several decades to stimulate other interesting insights on the linkage between fiscal deficit and economic growth in SSA countries.

The present study has certain limitations that are natural to any scientific investigation. First, the sample included only 42 countries across SSA countries. Future studies might contemplate including a larger number of countries to investigate the link between fiscal deficit and economic growth. Second, our study comprised a limited number of variables related to fiscal deficit and economic growth.

## Author Contributions

**Conceptualization:** Isubalew Daba Ayana.

**Data curation:** Isubalew Daba Ayana.

**Formal analysis:** Isubalew Daba Ayana.

**Methodology:** Isubalew Daba Ayana.

**Software:** Isubalew Daba Ayana.

**Supervision:** Atnafu Gebremeskel Sore, Isubalew Daba Ayana, Wondaferahu Mulugeta Demissie.

**Validation:** Isubalew Daba Ayana.

**Visualization:** Isubalew Daba Ayana.

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
