## [Decision Letter · Decision Letter 0]

1 Sep 2023

PONE-D-23-23990Effects of Fiscal deficit on economic growth in Sub-Saharan Africa: Fresh Evidence from System GMMPLOS ONE

Dear Dr. Ayana,

Thank you for submitting your manuscript to PLOS ONE. After careful consideration, we feel that it has merit but does not fully meet PLOS ONE’s publication criteria as it currently stands. Therefore, we invite you to submit a revised version of the manuscript that addresses the points raised during the review process.

We look forward to receiving your revised manuscript.

Kind regards,

Eric Fosu Oteng-Abayie

Academic Editor

PLOS ONE

2. Please amend the manuscript submission data (via Edit Submission) to include author Atnafu Gebremeskel.

Reviewers' comments:

Reviewer's Responses to Questions

**Comments to the Author**

1. Is the manuscript technically sound, and do the data support the conclusions?

Reviewer #1: Partly

Reviewer #2: Yes

2. Has the statistical analysis been performed appropriately and rigorously? 

Reviewer #1: Yes

Reviewer #2: I Don't Know

3. Have the authors made all data underlying the findings in their manuscript fully available?

Reviewer #1: No

Reviewer #2: Yes

4. Is the manuscript presented in an intelligible fashion and written in standard English?

Reviewer #1: No

Reviewer #2: No

5. Review Comments to the Author

Reviewer #1: Comments:

1. I have some reservations with the topic. Why not “Effects of fiscal deficit on economic growth in sub-Sahara Africa: A new evidence from system GMM”. I am not comfortable with the word “fresh”.

2. The effects of fiscal deficit on economic growth is overly researched even before the advent of the COVID-19 pandemic so what different or novelty is the study bringing on board. In fact, during the financial crisis in 2008 fiscal deficit became the issue in town. The study’s assertion in page 3 paragraph 2 that literature on impacts of fiscal deficits and economic growth is limited is not true and it defeats the discussions from the introductory page to that paragraph because the introductory discussions mainly focused on works that have been done on the subject. So why that assertion?

3. The rational and the motivation need to explicitly stated. The study’s rational and motivation needs to be re-enforced because the topic is overly researched in the macroeconomic literature. If it is about the updated data and new methodology of system GMM which is not new as claimed by the author(s) that needs to be emphasized in the study.

4. Under section 3.1, the author mentioned that the study followed the endogenous growth model theory but that model was not discussed under the theoretical review.

5. Under Table 2 caption, “sample” should be “Sample”.

6. Under section 3.2, the discussions on endogenous growth model should be done under the theoretical review and the author(s) should make references to it when specifying the model under section 3.2.

7. To be fair, the methodology used is appropriate and the pre-estimation test were adequately provided. However, the model specification is problematic. The study specifies a growth model and as such, should include other factors/variables that are known to impact economic growth and not only those chosen. As it stands, the model is misspecified. In recent times, literature on economic growth have found institutional and governance variables to impact economic growth and I expect the author(s) to include some of them.

8. The author(s) needs to provide substantiated reasons for the findings. It is not just presenting what is in the tables but adding economic explanations for these findings.

See attachment

Reviewer #2: 1. The write-up must be proofread. In its current state, the paper is difficult to read and appreciate its contributions.

2. Contribution - Authors stated (see page 3) that one of their contributions to the debate is the methodology employed but they failed to clearly state the weakness in the PMG methodology (used by Awolaja & Esefo, 2020) which GMM is addressing or why GMM results are more reliable/superior to PMG

3. The empirical literature review was poorly carried out as it generally reported the findings of previous studies without any form of discussion.

4. The authors transformed all variables to their natural logarithm including fiscal deficit. Meanwhile, authors computed fiscal deficit as the difference between tax revenue (% of GDP) and government final consumption expenditure (% of GDP) which suggests that there would be a lot of negative values yet authors proceeded to take the natural logarithm of these values (which is undefined) without any additional information

5. Other errors detected in the manuscript have been listed below:

• Authors interpret some I(0) variables as I(1) in Table 3 (page 16)

• On page 16 authors wrote “We are now confident enough that the result of our two-step system GMM is free from the risk of non-spurious regression.” but it’s not clear whether the authors meant spurious regression instead of non-spurious regression

• On page 18, the authors wrote “In addition to this, coefficient of real interest rate (logRIR) is found to have positive coefficient implying that a one percent changes in rate of real interest is associated with a 0.074 percent decline in economic growth, other things remaining constant.”, which is incorrect.

• On the same page (i.e. page 18), the authors stated “Thus, the result of the study is consistent across all estimated models.” but the real effective exchange rate and inflation variables have opposite signs in model 3 compared to model 1 and 2 (see Table 5).

• On page 18: “The result of the two-step system GMM revealed that a percentage change in the rate of inflation (consumer price index), all other things kept constant, is associated with a -0.035 decline in economic growth of SSA during 2011-2021. This result remained consistent across all the three columns of table 5.” but inflation has a positive sign in column 3.

6. Overall, the paper has not convincingly demonstrated the relevance of its contribution to the fiscal deficit and growth debate. Apart from the errors listed above, there are also too many typos, grammatical errors, and poor sentence constructions which may reduce the interest of the reader. The paper needs to be significantly improved to merit publication in PLOS ONE.

6. PLOS authors have the option to publish the peer review history of their article (what does this mean?). If published, this will include your full peer review and any attached files.

Reviewer #1: **Yes: **Eric Amoo Bondzie

Reviewer #2: No

---

## [Author Response · Author response to Decision Letter 0]

26 Sep 2023

Date: 9th September, 2023

Response to Reviewers’ Comments

Manuscript No.: PONE-D-23-23990

Title: Effects of Fiscal Deficit on Economic Growth in Sub-Saharan Africa: A New Evidence from System GMM

Journal: PLOS ONE

Dear Reviewers,

Greetings of the day!

We are thankful to the reviewers for taking the time to assess our manuscript, for their careful reading, and for their suggestions and valuable comments which helped us to substantially improve the quality of our paper. In revising the manuscript, we have carefully considered all the raised comments and suggestions. We have attempted to succinctly explain the changes made in reaction to all comments. Our reply to each comment in a point-by-point fashion is given in what follows

1. Regarding your concerns about the topic of the article 

Respected reviewers, thank you for your comment regarding the topic of our study. We have accepted it positively. In line with your comments, we have now changed the topic by avoiding the word Fresh. The corrected topic of the study is now: 

Effects of Fiscal Deficit on Economic Growth in Sub-Saharan Africa: A New Evidence from System GMM

2. On your concerns regarding introductory pages 

Dear Reviewers, thank you so much for your constructive comments regarding the introduction of our work. 

We have now corrected and articulated the introduction in a more articulated way. For instance, we have now solved the issue you raised regarding page 3 paragraph 2 of the original manuscript. 

Further, the paragraph that shows the organization of the study is included at the end of the introduction section. 

3. Regarding your concerns about the rationale and the motivation of the study

Dear reviewers, we would also like to thank you for your comment that improved our paper. 

The rationale and the motivation of the study are now explicitly explained (paragraphs 3 and 4 on page 3 of this manuscript). 

4. Regarding your concerns about the Literature Review

In line with your comments regarding the endogenous growth model in the theoretical literature review (see highlighted paragraphs on page 5 before section 2.2). 

5. On your concerns: Under Table 2 caption, “sample” should be “Sample”

Thank you very much, We have corrected editions and made a proof proofreads to substantially improve the paper. 

6. On Your concerns in section 3.2 of the manuscript 

Following your comments, we have made references to references to it when specifying the model under section 3.2. We have improved by making the connection between the literature section and the methodology section as per your comments. Thank you so much for your constructive comments. 

7. Regarding your concerns about model specification

Dear reviewers, we would like to thank you for your previous comments regarding the model. You have said: 

‘To be fair, the methodology used is appropriate and the pre-estimation test was adequately provided. ’ We are happy to hear this from you. 

We have substantially improved the model specification section in line with your comments. 

The main objective of our study is to examine the long and short-run effects of fiscal deficit in SSA countries. Thus, we have a fiscal deficit as a variable of interest. The rest are control variables. Our control variables included in the study are inflation (INF), real effective exchange rate (RER), real interest rate (IR), and gross fixed capital formation (GFCF). These variables are the fundamental problems of SSA countries. They are the major determinants of growth in Africa. They have direct impacts on growth. This is why we included them in our model. 

Since our objective is not to look at how governance and institutional qualities affect the linkage between fiscal deficit and economic growth, we haven’t included them in our study for now. This may be dubbed as the limitation of this study. Regardless of this, SSA countries’ policymakers want to know the short-run and long-run effects of fiscal deficit on economic growth. This is because growth is mainly financed by deficit and foreign borrowing. 

8. On your comments regarding discussions and findings 

 Dear reviewers, thank you so much for your comments once again. Frankly speaking, we have learned a lot and our paper has got a major benefit from your comments. 

Following your comments, we have interpreted our results substantially. We have provided reasons. We have included the result's economic meaning. It’s beyond mere interpretation of results from the table (We highlighted section 4 in red on pages 15 to 24). 

On your general Comments

1. On your suggestion regarding proofreading the manuscript

Following your comments, we have re-read our manuscripts and corrected grammatical errors in our work. We have also eliminated subject-verb agreement errors and inconsistencies in some sentences in our manuscript. Generally, we have seriously worked on this. 

2. Regarding your concerns about the conceptualization of the study 

 Dear reviewers, we have accepted your comments strongly and positively. 

The major concern of our study is to examine the long-run and short-run effects of fiscal deficits in SSA. Our study generates new evidence regarding this for the last ten years (second decades of the 21st century). Fiscal deficit was a major concern of literature during the global financial crises. Now, the world experienced a new event, a global pandemic that boosted fiscal deficit and inflation. It deteriorated foreign exchange and real interest rates. It seriously harassed global capital formation. Due to this, SSA countries are currently in a dilemma on whether to finance their growth using fiscal deficit or not. Thus, the examination of the long and short-run effects of fiscal deficit has magnificent importance for policymakers. 

 Moreover, please look at empirical literature in SSA countries. You can check the online. They are very limited. Contrary to this, look at the budget deficit of SSA countries. It is rising. The existing literature are reporting contradicting findings. Thus, Policymakers are in confusion. Thus, our current study has a vital contribution to the literature on SSA in this area. For that matter fiscal deficit is the fundamental macroeconomic problem of developing countries. As it adds a drop to the limited existing literature, we strongly believe that our current study has a significant contribution to the literature.

 On top of that in developing countries where government is more powerful, fiscal policy is the main tool of economic stabilization. A fiscal deficit is just the difference between government expenditure and government revenue. SSA countries are very much concerned about the difference in their income and expenditure. Thus, studying fiscal deficit is important for SSA literature. 

 Finally, we would like to thank once again all the reviewers for taking the time to review our manuscript, for their relevant remarks and comments, and especially for their specifications which helped us to improve the quality of our paper. 

 Sincerely yours, Isubalew Daba Ayana

---

## [Decision Letter · Decision Letter 1]

10 Nov 2023

PONE-D-23-23990R1Effects of Fiscal deficit on economic growth in Sub-Saharan Africa: A New Evidence from System GMMPLOS ONE

Dear Dr. Ayana,

Thank you for submitting your manuscript to PLOS ONE. After careful consideration, we feel that it has merit but does not fully meet PLOS ONE’s publication criteria as it currently stands. Therefore, we invite you to submit a revised version of the manuscript that addresses the points raised during the review process.

We look forward to receiving your revised manuscript.

Kind regards,

Eric Fosu Oteng-Abayie

Academic Editor

PLOS ONE

Additional Editor Comments:

I have considered the reviewer's comments and feel strongly that the authors did not sufficiently address them. The authors should note that the reviewer's comments are very significant in improving the paper for publication. Therefore, deep attention should be given to them. Authors should state any rebuttals, if necessary.

Reviewers' comments:

Reviewer's Responses to Questions

**Comments to the Author**

1. If the authors have adequately addressed your comments raised in a previous round of review and you feel that this manuscript is now acceptable for publication, you may indicate that here to bypass the “Comments to the Author” section, enter your conflict of interest statement in the “Confidential to Editor” section, and submit your "Accept" recommendation.

Reviewer #1: (No Response)

Reviewer #2: (No Response)

2. Is the manuscript technically sound, and do the data support the conclusions?

Reviewer #1: Yes

Reviewer #2: Partly

3. Has the statistical analysis been performed appropriately and rigorously? 

Reviewer #1: Yes

Reviewer #2: I Don't Know

4. Have the authors made all data underlying the findings in their manuscript fully available?

Reviewer #1: Yes

Reviewer #2: Yes

5. Is the manuscript presented in an intelligible fashion and written in standard English?

Reviewer #1: Yes

Reviewer #2: No

6. Review Comments to the Author

Reviewer #1: I still think the author(s) need to address the motivation of the paper well. Follow my earlier comment on the motivation.

"The rationale and the motivation need to be explicitly stated. The study’s rationale and motivation need to be reinforced because the topic is overly researched in the macroeconomic literature. If it is about the updated data and new methodology of system GMM, which is not new as claimed by the author(s), that needs to be emphasized in the study."

Reviewer #2: In my earlier comments, I raised some concerns to enhance the quality of the paper so as to merit publication in Plos One. However, upon reviewing the current version of the manuscript, I have noticed that the concerns I raised previously have not been addressed in the revised manuscript.

To clarify, I have listed my previous concerns (1-6) below:

1. The write-up needs proofreading. In its current state, the paper is difficult to read and appreciate its contributions.

2. Originality - Authors stated (see page 3) that one of their contributions to the debate is the methodology employed but they failed to clearly state the weakness in the PMG methodology (used by Awolaja & Esefo, 2020) which GMM is addressing or why GMM results are more reliable/superior to PMG

3. The empirical literature review was poorly carried out as it generally reported the findings of previous studies without any form of discussion.

4. The authors transformed all variables to their natural logarithm including fiscal deficit. Meanwhile, authors computed fiscal deficit as the difference between tax revenue (% of GDP) and government final consumption expenditure (% of GDP) which suggests that there would be a lot of negative values yet authors proceeded to take the natural logarithm of these values (which is undefined) without any additional information

5. Other errors detected in the manuscript have been listed below:

• Authors interpret some I(0) variables as I(1) in Table 3 (page 16)

• On page 16 authors wrote “We are now confident enough that the result of our two-step system GMM is free from the risk of non-spurious regression.” but it’s not clear whether the authors meant spurious regression instead of non-spurious regression

• On page 18, the authors wrote “In addition to this, coefficient of real interest rate (logRIR) is found to have positive coefficient implying that a one percent changes in rate of real interest is associated with a 0.074 percent decline in economic growth, other things remaining constant.”, which is incorrect.

• On the same page (i.e. page 18), the authors stated “Thus, the result of the study is consistent across all estimated models.” but the real effective exchange rate and inflation variables have opposite signs in model 3 compared to model 1 and 2 (see Table 5).

• On page 18: “The result of the two-step system GMM revealed that a percentage change in the rate of inflation (consumer price index), all other things kept constant, is associated with a -0.035 decline in economic growth of SSA during 2011-2021. This result remained consistent across all the three columns of table 5.” but inflation has a positive sign in column 3.

6. Overall, the paper has not convincingly demonstrated the relevance of its contribution to the fiscal deficit and growth debate. Apart from the errors listed above, there are also too many typos, grammatical errors, and poor sentence constructions which may reduce the interest of the reader. The paper needs to be significantly improved to merit publication in PLOS ONE.

Although the authors may have attempted to proofread the manuscript, there is still room for improvement in terms of language quality to enhance clarity and reader engagement. It is worth noting that the authors have not addressed the remaining issues (numbered 2-5) and have also failed to provide any rebuttals. Some of these issues are crucial in understanding the paper's contribution and evaluating the accuracy and dependability of the findings.

7. PLOS authors have the option to publish the peer review history of their article (what does this mean?). If published, this will include your full peer review and any attached files.

Reviewer #1: No

Reviewer #2: No

---

## [Author Response · Author response to Decision Letter 1]

14 Nov 2023

Date: 14th November, 2023

Response to Reviewers’ Comments

Manuscript No.: PONE-D-23-23990R1

Title: Effects of Fiscal Deficit on Economic Growth in Sub-Saharan Africa: A New Evidence from System GMM

Journal: PLOS ONE

Dear Reviewers,

Greetings of the day!

We are thankful to the reviewers for taking the time to assess our manuscript, for their careful reading, and for their suggestions and valuable comments which helped us to substantially improve the quality of our paper. In revising the manuscript for the second time, we have carefully considered all the raised comments and suggestions. We have attempted to succinctly explain the changes made in reaction to all comments. Our reply to each comment in a point-by-point fashion is given in what follows

1. On comment 1: The write-up needs proofreading. In its current state, the paper is difficult to read, and appreciate its contributions.

Answers: 

Dear reviewers, thank you so much for your comments. We have accepted your comments strongly and positively. Following your comments, we have re-read our manuscripts to enhance clarity and reader engagement. We have eliminated too many typos, grammatical errors, and poor sentence constructions. We believe that your comments made our manuscript smarter than before. 

2. On comment 2: Originality - Authors stated (see page 3) that one of their contributions to the debate is the methodology employed but they failed to clearly state the weakness in the PMG methodology (used by Awolaja & Esefo, 2020) which GMM is addressing or why GMM results are more reliable/superior to PMG

Answer: 

Dear reviewers, we have accepted your comments and suggestions positively. We have now explained the major reason why system GMM is superior to PMG. ( page 3 last paragraph), the highlighted paragraph. Your constructive comments are well appreciated. 

3. On comment 3: The empirical literature review was poorly carried out as it generally reported the findings of previous studies without any form of discussion.

Answer: 

Dear reviewers, we are very grateful for your comments on the literature section of our manuscript. Following your good comments, we have written the literature section more attractively. We have linked the ideas. We have included a detailed discussion regarding the findings. It is now more beautiful with a comprehensive discussion. The literature portion is now enriched. For convenience, The entire changes made in the literature review section are highlighted. Thank you for your constructive comments. 

4. On comment 4:

The authors transformed all variables to their natural logarithm including fiscal deficit. Meanwhile, authors computed fiscal deficit as the difference between tax revenue (% of GDP) and government final consumption expenditure (% of GDP) which suggests that there would be a lot of negative values yet authors proceeded to take the natural logarithm of these values (which is undefined) without any additional information

Answer 

Dear reviewers thank you once again for raising another essential question that improved our manuscript. Sure, the fiscal deficit variable is the difference between government revenues and government expenditures. We have faced some negative values. It makes no sense when we convert the negative value to a natural log. However, we have handled this problem scientifically. Page 15 above section 3.3 has a solution for this. It is written like this. Before transforming the value of a major explanatory variable, fiscal deficit, we faced some negative values of fiscal deficit as a percentage of GDP when we calculated the difference between government revenue and government expenditure. Since the log of the negative number is none in economics, as it is in mathematics, we used the most common technique of transforming to handle the negative values in our fiscal deficit data set. To do this, we added a constant value to our fiscal deficit data set before applying the log transform. This method is preferred as it is popular, easy, and more appropriate for managing a few negative numbers that appear in the data. Now, it is meaningful. Thank you for your comment that made our manuscript smarter. 

5. On Comment 5: Other errors detected in the manuscript have been listed below: • Authors interpret some I(0) variables as I(1) in Table 3 (page 16) • On page 16 authors wrote “We are now confident enough that the result of our two-step system GMM is free from the risk of non-spurious regression.” but it’s not clear whether the authors meant spurious regression instead of non-spurious regression • On page 18, the authors wrote “In addition to this, coefficient of real interest rate (logRIR) is found to have positive coefficient implying that a one percent changes in rate of real interest is associated with a 0.074 percent decline in economic growth, other things remaining constant.”, which is incorrect. • On the same page (i.e. page 18), the authors stated “Thus, the result of the study is consistent across all estimated models.” but the real effective exchange rate and inflation variables have opposite signs in model 3 compared to model 1 and 2 (see Table 5). • On page 18: “The result of the two-step system GMM revealed that a percentage change in the rate of inflation (consumer price index), all other things kept constant, is associated with a -0.035 decline in economic growth of SSA during 2011-2021. This result remained consistent across all the three columns of table 5.” but inflation has a positive sign in column 3.

Answer: 

Dear reviewers, we would like to thank you for your time again. You have given a lot of detailed comments under comments 5. Many are related to Table 3 and Table 5. We have looked carefully into them and corrected all of them. We have highlighted them carefully. In Table 3( page 19 of the revised manuscript), we carried out stationarity at the level and first difference. The result shows that all variables are stationary at first difference. Some variables are not stationary at the level. For system GMM, we need data to be stationary. It might be either at level or first difference. Variables that are not stationary at the level are stationary at the first difference. The corresponding decisions were provided at both I(0) and I(1) in both unit root test cases. Thus, the major concern here is solved. 

Finally, we would like to thank once again all the reviewers for taking the time to review our manuscript, for their relevant remarks and comments, and especially for their specifications which helped us to improve the quality of our paper. 

 Sincerely yours, Isubalew Daba Ayana

---

## [Decision Letter · Decision Letter 2]

15 Dec 2023

PONE-D-23-23990R2Effects of Fiscal deficit on economic growth in Sub-Saharan Africa: A New Evidence from System GMMPLOS ONE

Dear Dr. Ayana,

Thank you for submitting your manuscript to PLOS ONE. After careful consideration, we feel that it has merit but does not fully meet PLOS ONE’s publication criteria as it currently stands. Therefore, we invite you to submit a revised version of the manuscript that addresses the points raised during the review process.

We look forward to receiving your revised manuscript.

Kind regards,

Eric Fosu Oteng-Abayie

Academic Editor

PLOS ONE

Journal Requirements:

Additional Editor Comments :

Note: Please take your time to sufficiently address the reviewer's comments and also proofread the manuscript to improve the flow of language. This is to avoid reviewer fatigue.

Reviewers' comments:

Reviewer's Responses to Questions

**Comments to the Author**

1. If the authors have adequately addressed your comments raised in a previous round of review and you feel that this manuscript is now acceptable for publication, you may indicate that here to bypass the “Comments to the Author” section, enter your conflict of interest statement in the “Confidential to Editor” section, and submit your "Accept" recommendation.

Reviewer #1: (No Response)

2. Is the manuscript technically sound, and do the data support the conclusions?

Reviewer #1: Partly

3. Has the statistical analysis been performed appropriately and rigorously? 

Reviewer #1: Yes

4. Have the authors made all data underlying the findings in their manuscript fully available?

Reviewer #1: Yes

5. Is the manuscript presented in an intelligible fashion and written in standard English?

Reviewer #1: No

6. Review Comments to the Author

Reviewer #1: To be fair, the author(s) need to address the following issues:

1. The rationale for the study is not captivating. Just like I said before the author(s) need to still motivate the work.

2. The grammatical errors in the paper need to be looked at. There are so many subject-verb agreement errors and inconsistencies in some of the sentences. I advised earlier for the service of a proofreader and I still think the author(s) need that.

7. PLOS authors have the option to publish the peer review history of their article (what does this mean?). If published, this will include your full peer review and any attached files.

Reviewer #1: No

---

## [Author Response · Author response to Decision Letter 2]

4 Mar 2024

Date: 2nd March, 2024

Response to Reviewers’ Comments

Manuscript No.: PONE-D-23-23990R2

Title: On the Fiscal Deficit and Economic Growth in Sub-Saharan Africa: A New Evidence from System GMM 

Journal: PLOS ONE

Greetings, Reviewers 

Salutations for the day! 

We appreciate the reviewers' time in evaluating our article, their attentive reading, and their insightful recommendations and remarks, which enabled us to significantly raise the caliber of our work. The second revision of the article has taken into account all of the pointed criticisms and recommendations. We have tried to provide a clear explanation of the modifications made in response to all feedback. What follows is a point-by-point summary of our response to each comment.

1. On comment 1: 1. The rationale for the study is not captivating. Just like I said before the author(s) need to still motivate the work.

Answers: 

Greetings, reviewers Your thoughts and opinions have been positively received by us. We appreciate your feedback, which made our job better. We have now stated the study's goal in clearer terms. View the paragraphs on page 3's end.

2. On comment 2: The grammatical errors in the paper need to be looked at. There are so many subject-verb agreement errors and inconsistencies in some of the sentences. I advised earlier for the service of a proofreader and I still think the author(s) need that.

Answer: 

Greetings, reviewers We value the helpful criticism you have provided. The linguistic editing is done with great care.

---

## [Decision Letter · Decision Letter 3]

2 May 2024

On the Fiscal Deficit and Economic Growth in Sub-Saharan Africa: A New Evidence from System GMM

PONE-D-23-23990R3

Dear Dr. Ayana,

We’re pleased to inform you that your manuscript has been judged scientifically suitable for publication and will be formally accepted for publication once it meets all outstanding technical requirements.

Kind regards,

Eric Fosu Oteng-Abayie

Academic Editor

PLOS ONE

Additional Editor Comments (optional):

The author has sufficiently addressed the reviewer’s comments. The paper now satisfies the publication criteria of the journal.

Reviewers' comments:

Reviewer's Responses to Questions

**Comments to the Author**

1. If the authors have adequately addressed your comments raised in a previous round of review and you feel that this manuscript is now acceptable for publication, you may indicate that here to bypass the “Comments to the Author” section, enter your conflict of interest statement in the “Confidential to Editor” section, and submit your "Accept" recommendation.

Reviewer #1: All comments have been addressed

2. Is the manuscript technically sound, and do the data support the conclusions?

Reviewer #1: Yes

3. Has the statistical analysis been performed appropriately and rigorously? 

Reviewer #1: Yes

4. Have the authors made all data underlying the findings in their manuscript fully available?

Reviewer #1: Yes

5. Is the manuscript presented in an intelligible fashion and written in standard English?

Reviewer #1: Yes

6. Review Comments to the Author

Reviewer #1: General Comments

1. The rationale for the study has been motivated.

2. The grammatical errors in the paper has been looked at.

7. PLOS authors have the option to publish the peer review history of their article (what does this mean?). If published, this will include your full peer review and any attached files.

Reviewer #1: No

---

## [Editor Report · Acceptance letter]

12 Jul 2024

PONE-D-23-23990R3 

PLOS ONE

Dear Dr. Ayana, 

I'm pleased to inform you that your manuscript has been deemed suitable for publication in PLOS ONE. Congratulations! Your manuscript is now being handed over to our production team.

Kind regards, 

on behalf of

Dr. Eric Fosu Oteng-Abayie 

Academic Editor

PLOS ONE